# *Trackoscope*: A low-cost, open, autonomous tracking microscope for long-term observations of microscale organisms

**Priya Soneji**[1], **Elio J. Challita**[1], **Saad Bhamla**[2]*

**1** George W. Woodruff School of Mechanical Engineering, Georgia Institute of Technology, Atlanta, GA, United States of America, **2** School of Chemical and Biomolecular Engineering, Georgia Institute of Technology, Atlanta, GA, United States of America

* saadb@chbe.gatech.edu

**Data Availability Statement:** All the data required to replicate the results are included within the Supporting information + github + figshare: a. https://github.com/bhamla-lab/Trackoscope b.

## Abstract

Cells and microorganisms are motile, yet the stationary nature of conventional microscopes impedes comprehensive, long-term behavioral and biomechanical analysis. The limitations are twofold: a narrow focus permits high-resolution imaging but sacrifices the broader context of organism behavior, while a wider focus compromises microscopic detail. This trade-off is especially problematic when investigating rapidly motile ciliates, which often have to be confined to small volumes between coverslips affecting their natural behavior. To address this challenge, we introduce *Trackoscope*, a 2-axis autonomous tracking microscope designed to follow swimming organisms ranging from $10\mu m$ to $2mm$ across a $325cm^2$ area (equivalent to an A5 sheet) for extended durations—ranging from hours to days—at high resolution. Utilizing *Trackoscope*, we captured a diverse array of behaviors, from the air-water swimming locomotion of *Amoeba* to bacterial hunting dynamics in *Actinosphaerium*, walking gait in *Tardigrada*, and binary fission in motile *Blepharisma*. *Trackoscope* is a cost-effective solution well-suited for diverse settings, from high school labs to resource-constrained research environments. Its capability to capture diverse behaviors in larger, more realistic ecosystems extends our understanding of the physics of living systems. The low-cost, open architecture democratizes scientific discovery, offering a dynamic window into the lives of previously inaccessible small aquatic organisms.

## Beyond conventional microscopy: Enabling the study of microorganism motility

Microscopy serves as a pivotal tool for delving into the microscopic realm, facilitating the study of the inner mechanisms and behaviors of organisms. Traditional microscopy, constrained by a fixed lens, falls short in capturing the full spectrum of microorganism motility. The manual tracking of these microscale entities under a microscope presents challenges due to their diverse sizes, velocities, and the precision required for observation. A tracking microscope mitigates these issues, enabling precise and efficient monitoring of an organism's

https://figshare.com/projects/Trackoscope_A_
Low-Cost_Open_Autonomous_Tracking_
Microscope_for_Long-Term_Observations_of_
Microscale_Organisms/200934.

**Funding:** NIH Grant R35GM142588; NIGMS SEPA
Grant R25GM142044; NSF Grants MCB-1817334;
CAREER IOS-1941933; and the Open Philanthropy
Project The funders had no role in study design,
data collection and analysis, decision to publish, or
preparation of the manuscript.

**Competing interests:** The authors have declared
that no competing interests exist.

movement, thus fostering more controlled experimentation and detailed observation of behavioral patterns and microstructures. It automates the tracking process, conserving time, reducing user error, and minimizing variability.

Several tracking solutions exist within the microscopy landscape, including motorized stage microscopes, laser tracking systems, and image-based tracking systems. Motorized stage microscopes utilize motors to maneuver the stage that holds the sample, facilitating user control over sample movement with autonomous tracking capabilities within a specified area (up to 144 $cm^2$) or infinitely in one plane [1–3]. Laser tracking systems, employing lasers to detect and follow sample movement, adjust the microscope's focus accordingly [4]. Image-based tracking systems, prevalent in the field, apply computer vision algorithms to follow the sample's movement through images captured by the microscope [5–8]. While these methods mark significant strides in data collection for organism study, they are not without drawbacks. Image-based systems are bound by the resolution and field of view of the imaging device, laser systems are adept at following single particles but are less effective with complex organisms, and motorized stage microscopes, while versatile, can be costly and restricted by the size of the trackable area. Additionally many of these setups can be quite expensive, costing between 1000 and 5000 dollars [2, 3].

Despite recent advancements in accessible, affordable, and open hardware microscopy [9–15], the ability to observe moving organisms remains a significant challenge with traditional microscopes. To bridge this gap, we introduce *Trackoscope*, a low-cost, open tracking microscope. *Trackoscope*, costing around 400 dollars in parts, is accessible when compared to traditional light-field microscopes and can be assembled by a minimally trained individual. It employs image-based tracking to autonomously follow and focus on moving organisms, enabling precise tracking over an expansive field of view (approximately 325 $cm^2$). Equipped with a 12-megapixel camera, *Trackoscope* captures video data amenable to machine-learning-based behavior analysis pipelines. The affordability, customizability, and ease of assembly make *Trackoscope* an invaluable asset in both university-level research and K-12 education, fostering the exploration and analysis of micro-organism behaviors.

## *Trackoscope*: An affordable microscope with open hardware and software

### *Trackoscope*'s low-cost motorized XY stage

*Trackoscope* is a modular, lead screw-driven, two-axis actuator designed for automated visual tracking of motile microorganisms. It can be controlled by a standard computer or laptop using an Arduino Uno and communicates with the computer via a USB-based serial communication (COM) port (Fig 1). The driver stack can support any motorized stages that are stepper motor-based. The *Trackoscope* design features motorized X and Y stages with an 18 cm x 18 cm travel range, powered by two NEMA 17 stepper motors with 400 steps/rev (precision of 0.9 degrees), which are controlled by a CNC Shield (S1 Fig). The CNC Shield allows for the increase of torque and the addition of micro-stepping, which can increase precision up to 32 times. Specifically, micro-stepping allows the same motors to move at a range of speeds, from 4,600 $\mu m/s$ for fast-swimming ciliates to 145 $\mu m/s$ for slow-moving amoeba and tardigrades.

To improve precision and absorb vibrations from potential assembly misalignments, the X and Y axes have pulleys with a 3:1 gear ratio. The XY stage utilizes a stacked design, with the X axis driving the Y axis, which then moves a magnetically-attached optics stage. This stacked system simplifies the assembly since each axis can be constructed separately, and it also allows for modularity if movement in only one axis is needed. Based on observations that a thin liquid

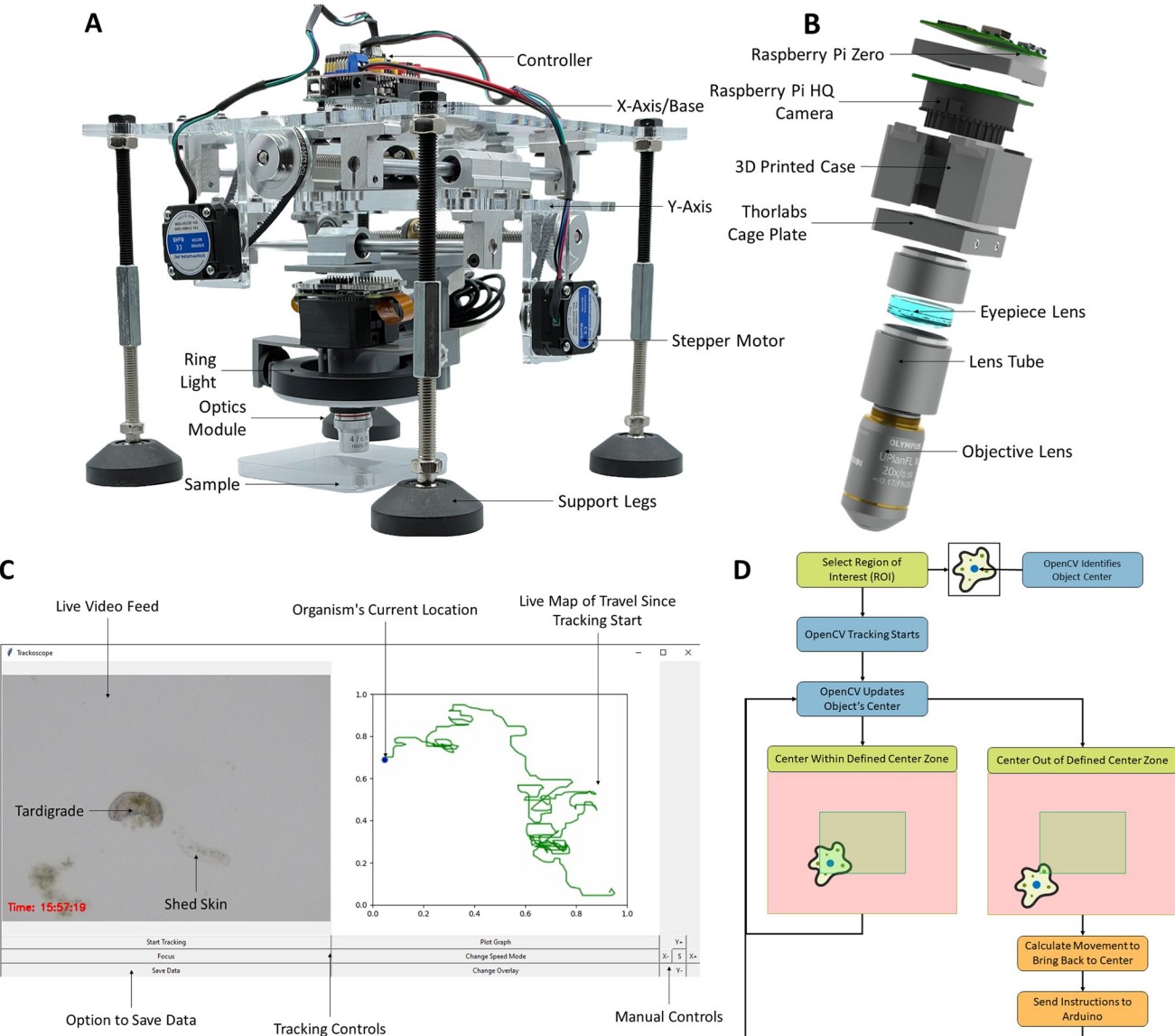

**Fig 1. Comprehensive view of *Trackoscope*: Design, interface, and tracking mechanics. (A)** The *Trackoscope* prototype in the inverted tracking position. The platform's footprint is 900 $cm^2$, with a tracking area of 325 $cm^2$. **(B)** Displays a CAD model of the optics assembly. The objective lenses are interchangeable, permitting the observation of a variety of organisms. The Raspberry Pi HQ Camera alongside the Raspberry Pi Zero function as a webcam, allowing this custom digital microscope to interface with any device. **(C)** Demonstrates the *Trackoscope* user interface, featuring a live video feed, real-time tracking map, manual actuator controls, and data saving options. **(D)** Illustrates the flowchart of the tracking algorithm and the visualization of logic. Utilizing OpenCV's built-in tracking, the algorithm pinpoints the organism's location based on the user's initial selection of a region of interest. The organism's location informs the actuator's movements to maintain the organism within the field of view.

layer in the organism sample negates the need for Z-axis adjustments, we opted for a manual Z-stage in this design. This choice reduces cost and simplifies user modifications to the optics unit. Additionally, *Trackoscope* can also be inverted to improve imaging based on the organism (S2 Fig).

The footprint of *Trackoscope* is 900 $cm^2$ (fits within a square foot), weighs 3.4 kg (equivalent to a gallon of milk), and the actuator itself costs $167 (Table 1).

**Table 1. Detailed component costs for assembling *Trackoscope*.**

| Part | Description | Cost |
|---|---|---|
| Arduino Uno + CNC Shield | Shield Expansion Board V3.0, R3 Board, A4988 Stepper Motor Driver | $28.88 |
| NEMA 17 Stepper Motors | Nema 17 Bipolar 0.9deg 11Ncm 42x42x21mm | $25.08 |
| Threaded Rods Set | 200mm Horizontal Optical Axis and 8mm Lead Screw Dual Rail Shaft Support, Pillow Block Bearings | $56.58 |
| Belt + Pulley | PGT2 Aluminum Timing Belt Idler Pulley Bearing 20/60 Teeth | $21.98 |
| Legs | M10 Thread Adjustable Foot Cups, M10 Hex Coupling Nuts, 100mm Steel Hex Head Screws | $19.53 |
| Lenses | Ø1" Achromatic Doublet and 4X Achromatic Microscope Objective | $125.36 |
| Ring Light | 4" Ring Light for Laptop | $11.99 |
| Raspberry Pi Camera | Raspberry Pi High Quality HQ Camera—12MP | $50 |
| Lens Tubes | SM1 Lens Tube, 0.50" Thread Depth and SM1 Lens Tube, 1.00" Thread Depth | $29.03 |
| Raspberry Pi | Raspberry Pi Zero 2 W | $15 |
| Cage Plate | 30 mm Cage Plate with Ø1" Double Bore | $21.11 |
| 3D Printed Parts | 200 grams of PLA filament | $4 |
| MDF | 1/4 in. x 1 ft. x 2 ft. Medium Density Fiberboard | $5 |
| Fasteners | M3/M4/M5 Screws and Bolts | $10 |
| *Total Actuator Cost* | | *$167.05* |
| *Total Optics Cost* | | *$256.49* |
| **Total Cost** | | **$423.54** |

Note: This table lists the costs of individual components for assembling *Trackoscope*. For detailed product numbers (P/N) and where to purchase (links), refer to S1 Table in S1 File. All prices were accessed as of January 2023.

## The optical core of *Trackoscope*

The optics module consists of four different components: an imaging sensor, an achromatic doublet lens, a lens tube, and an objective (Fig 1b). The imaging sensor is a Raspberry Pi High-Quality Camera paired with a Raspberry Pi Zero. A bare sensor works best as it optimizes the largest possible field of view (FOV). While the Raspberry Pi High-Quality Camera (12.3 MP Sony IMX477 sensor, 1.55 um pixel size) is the most economical option at $50, higher quality cameras can also be attached to further increase imaging resolution. A higher-end $400 camera (Imaging Source DFK 37BUX273 USB 3.1) was also tested and produced images of similar quality to the Raspberry Pi camera. Beneath the camera is an achromatic doublet lens inside a lens tube, at the end of which a variety of objectives can be connected, ranging from 2X to 20X. With the distance of achromatic doublet within the lens tube, the compounded magnification of the microscope can range from 20X to 200X, depending on the camera sensor used, as the focal length of the sensor affects the initial magnification. A LED ring light around the optics provides bright field illumination, and a white backing is placed behind the sample to provide a clean background. Dark field illumination can also be used with *Trackoscope* by adding a 3D printed attachment to the ring light that directs the light around the sample from below.

## Micro-tracking with OpenCV

To track the organisms, we employ OpenCV's Channel and Spatial Reliability Tracker (CSRT) [16, 17]. The CSRT is favored over machine learning trackers for its broader applicability

without the need for a high-performance GPU. By allowing the user to select the organism or region of interest (ROI), the tracker bypasses the necessity of pre-training with the specific organism being tracked. In the graphical user interface (GUI) (Fig 1c), designed using Tkinter, the user can view a live video feed from the microscope, observe the live trajectory of the organism, manually operate the *Trackoscope*, and save tracking data as a CSV file.

Upon selecting an organism for tracking, the user clicks "start tracking" in the GUI, prompting a window for manual organism selection using the cursor. After selection and hitting the "enter" key, tracking commences. OpenCV identifies the organism and its bounding box from the user's initial selection, then continually updates the box position as the organism moves. The CSRT model updates the bounding box by continually adapting over the duration of the track as it considers both the appearance of the organism and its motion characteristics. This adaptability makes the CSRT particularly effective in handling challenges such as occlusion, rotation, and scale variation. The bounding box data then informs actuator movement calculations (Fig 1d). If the bounding box center moves outside the central zone, commands are issued to the Arduino to adjust the axis until the organism is re-centered in the video frame. This feedback mechanism ensures the organism remains within the microscope's field of view (FOV). The use of a central zone instead of an exact center optimizes the smoothness of the video by reducing the chance of overcompensating movements, especially with organisms that may stop or pivot abruptly. Additionally, as the CSRT model evolves over the track, the perceived center of the organism may not always be accurate or precise; this is another factor for the incorporation of a central zone as the perceived center, while always on the organism, may not be on the exact center.

## Intuitive and custom graphical user interface

The user interface provides access to tools and data useful throughout the tracking process (Fig 1c). The GUI displays a live video feed from the digital microscope, averaging 107 FPS at a resolution of 640 x 480, which is adjustable. A timestamp is also included for reference during analysis. A live trajectory map on the right side of the video feed plots points based on actuator movements during tracking, with a blue hexagon indicating the organism's current position. Tracking locations and timestamps are saved over time in a CSV file for later download. Manual actuator controls are available in the GUI's bottom right along with various tracking-related controls such as toggling telemetry data display or stage smoothness adjustments (S1 Video).

## Performance evaluation of tracking and imaging capabilities

We evaluated the *Trackoscope*'s resolution capabilities, lighting, and tracking speed (Fig 2). Resolution was empirically determined using USAF 1951 Resolution Targets (Thorlabs R1DS1P), with the 4x objective at 35.08 $\mu m$ (Fig 2d) and the 10x objective at 8.77 $\mu m$ (Fig 2e). The resolution formula for the USAF 1951 target is $RES_{LP} = 2^{Group+(Element-1)/6}$, with the center-to-center line distance given by $RES_{CC} = 1000/RES_{LP}$. This resolution allows for the observation of detailed organism features such as cilia clusters or internal structures (Fig 2b and S2 Video). Dark field imaging allows for higher contrast when observing more translucent organisms (Fig 2c and S2 Video). Image brightness was assessed for the Raspberry Pi HQ microscope setup using the Python Image Library (PIL) to analyze grayscale pixel brightness. Brightness levels were 88.32% for the Amscope ring light and 58.21% for the AIXPI ring light. Speed was measured by timing a 1 cm travel at different micro-stepping settings (Fig 2f). Depending on the organism's speed, an appropriate micro-stepping setting is selected, typically twice the organism's speed.

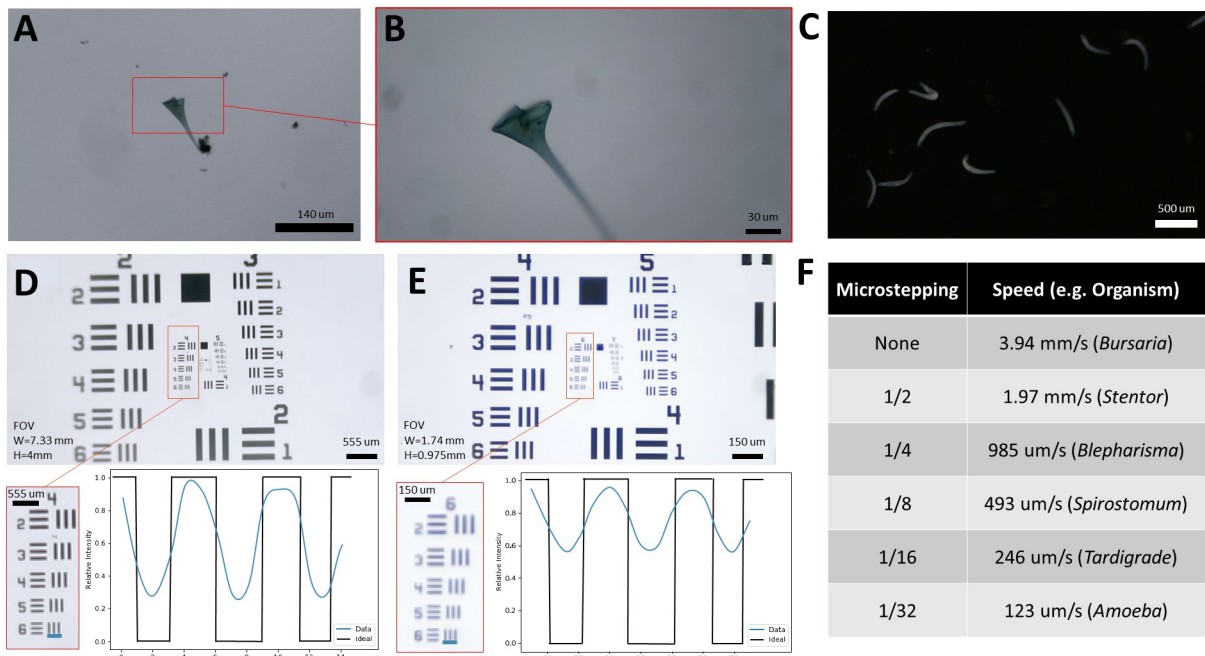

**Fig 2. Resolution and speed profiling of *Trackoscope*.** (A) Microscope image resolution of *Stentor coeruleus* with a 4x objective lens. (B) Detail resolution of *Stentor* with a 10x objective lens. (C) Dark field image of *Spirostomum ambiguum* with a 4x objective lens. (D) Image of a USAF 1951 resolution target captured with a 4X objective on the Raspberry Pi HQ microscope setup, showing an enlarged view of Group 4 and an intensity profile along the indicated blue line. This confirms resolution for Group 4, Element 6, equivalent to 28.51 Line Pairs/mm or a 35.08 $\mu$m resolution. (E) Image of a USAF 1951 resolution target with a 10x objective on the Raspberry Pi HQ microscope setup, showing an enlarged view of Group 6 and an intensity profile along the indicated blue line. This confirms resolution for Group 6, Element 6, equivalent to 114.04 Line Pairs/mm or an 8.77 $\mu$m resolution. (F) Chart of platform speeds across various micro-stepping settings, with examples of trackable organisms at those speeds.

## Movement pattern analysis from tracking data

We analyze the position and video tracking data to extract movement patterns, speeds, feeding behaviors, and other organism dynamics. Data points are recorded every 50 ms, capturing the timestamp, the organism's position relative to the center of the video frame ($x_{o/f}$, $y_{o/f}$), and the platform's position ($x_f$, $y_f$) relative to the starting point. The organism's position $P(x_o, y_o)$ is computed by adding the displacements of the platform and the organism's position within the video frame: $P(x_o, y_o) = P(\Delta x_{o/f} + \Delta x_p, \Delta y_{o/f} + \Delta y_p)$ (S3 Fig).

This displacement data facilitates the calculation of organism speed and identification of features such as feeding-swimming transitions and gait-switching frequencies. During tracking, we also utilize Open Broadcast Studio [18] to record the tracking window, providing visual data for subsequent analysis of shape changes and gait through visual tracking.

## Microscopic tracking of varied organism behaviors

We demonstrate the capabilities of *Trackoscope* for tracking microscopy by analyzing the movement of organisms varying in size, speed, and behavior. This tracking not only showcases the range of *Trackoscope*'s functionality but also provides insights into unique organism behaviors.

**High-speed ciliate tracking.** We tracked fast-moving ciliates such as *Bursaria truncatella* (800 $\mu$m), *Blepharisma japonicum* (500 $\mu$m), *Spirostomum ambiguum* (600 $\mu$m), and *Stentor coeruleus* (600–900 $\mu$m). The long-duration tracks of *Bursaria truncatella* and *Stentor coeruleus*

test *Trackoscope*'s ability to maintain focus on rapidly moving ciliates, which typically escape the view of traditional microscopes instantaneously. For instance, *Bursaria truncatella*, shaped like a scoop, achieves top speeds of 1775 *μm/s* (11.8 body lengths/second) multiple times within an 18-minute track, covering a distance of 45 centimeters (Fig 3a). The mutualistic endosymbiotic relationship between *Bursaria truncatella* and green algae (*Chlorella*) triggers faster movements under light [19], a condition enhanced by *Trackoscope*'s light microscopy setup, allowing *Bursaria truncatella* to reach peak speeds. Manually tracking such swift organisms without *Trackoscope* would be nearly impossible for extended periods.

Additionally, we monitored *Blepharisma* for up to 1.5 hours, observing speeds up to 500 *μm/s* (2 body lengths/second) (Fig 3c). We successfully tracked the entire process of asexual reproduction by binary fission in *Blepharisma*, including cytokinesis over 78 minutes (S3 Video) (Fig 4d).

*Spirostomum ambiguum* exhibited ciliary movement along with multiple contractions, maintaining an average speed of 161 microns/second and demonstrating the most consistent speed distribution among the observed organisms (Fig 3b and 3d).

Tracking of *Stentor coeruleus* provided a window into predator-prey interactions and morphological changes during different locomotive states (Fig 4a). As it moves, *Stentor coeruleus* adopts a more spherical shape, and when feeding, it anchors itself with a holdfast and assumes its characteristic trumpet shape [20]. The *Trackoscope*'s high-resolution video capture enables clear visualization of feeding behaviors using the cilia clusters in its oral pouch (S2 Video). These observations are informed by the long-term tracking of *Stentor coeruleus* that *Trackoscope* enables. This extended tracking empowers researchers to observe subtle behavioral patterns, shedding light on potential mechanisms governing information processing, learning, and regeneration in these simple-celled organisms.

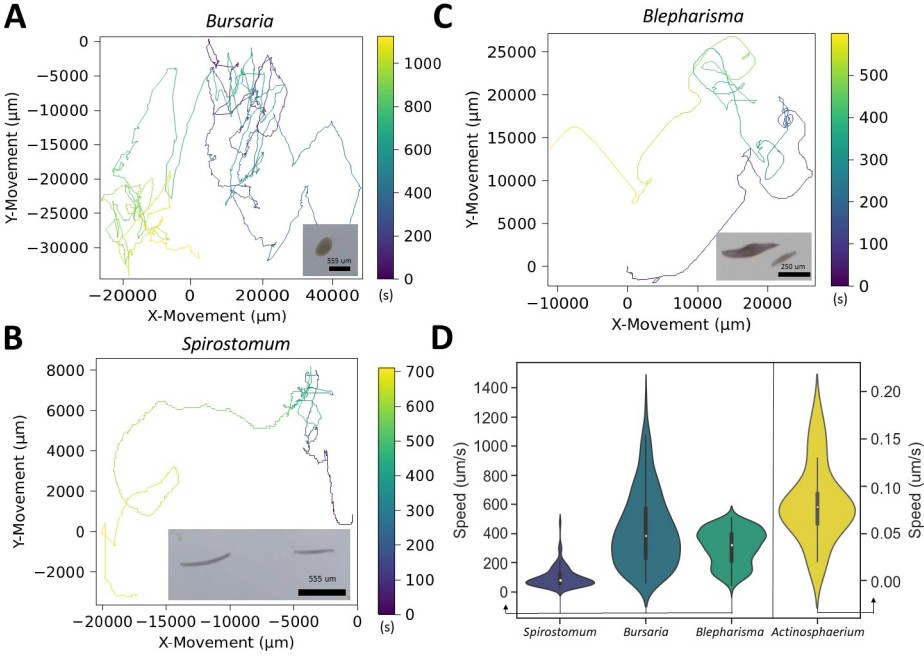

**Fig 3. *Trackoscope*'s versatility in speed adaptation: Profiling rapid and slow microorganism movement. (A)** *Bursaria truncatella* tracked over 18 minutes. **(B)** *Spirostomum ambiguum* tracked over 12 minutes. **(C)** *Blepharisma* tracked over 10 minutes. **(D)** The violin plot shows the speed distributions of *Spirostomum ambiguum*, *Bursaria truncatella*, *Blepharisma*, and *Actinosphaerium*, illustrating the broad spectrum of speeds *Trackoscope* can handle.

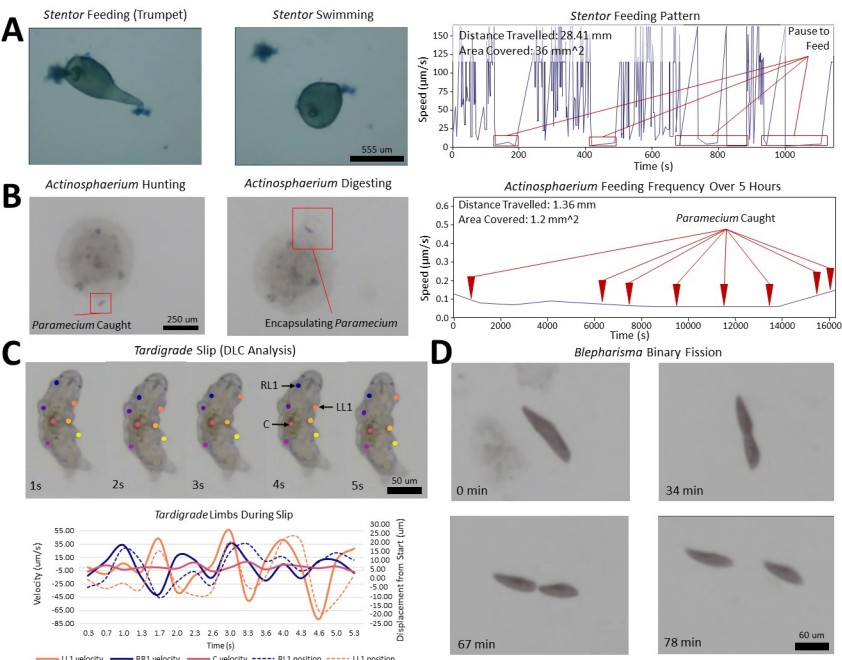

**Fig 4. Behavioral diversity in microscopy: *Trackoscope*'s wide-ranging organism tracking capabilities. (A)** *Stentor coeruleus* assumes different geometries as it feeds (trumpet shape) and swims (spherical shape) over 25 minutes. While in the feeding position, different anatomical features such as the holdfast and oral pouch are also visible (S3 Video). **(B)** *Actinosphaerium* hunting *Paramecium* over 5.5 hours (S3 Video). Features such as axopods and the development of contractile vacuoles for digesting are visible throughout the track. **(C)** Deeplabcut analysis of *Tardigrada* as it slips on a petri dish over five seconds. The velocities and positions of the front two legs and mass center of the *Tardigrada* over the five seconds. **(D)** Binary fission of motile *Blepharisma* over 1.5 hours (S3 Video).

**Observing slow-mover behaviors.** For slower unicellular organisms, we track *Actinosphaerium eichhorni* (500 µm) rolling and drifting with water currents and *Amoeba proteus* (450 µm) crawling. *Actinosphaerium*, with its sea urchin-like shape and numerous extending axopodia, reaches a maximum of 12 µm/min (0.02 body lengths/minute) as it advances slowly [21]. Over 5.5 hours of tracking, we observe the capture and digestion of 8 *Paramecium caudatum* (Fig 4b). The SI video also reveals how *Actinosphaerium* employs its axopodia to seize *Paramecium* and contractile vacuoles for encapsulation and digestion [22].

By tracking *Amoeba proteus* for 1.5 hours, *Trackoscope* discloses new insights that augment existing research. We track the dual locomotion of *Amoeba proteus*, crawling on a confined glass slide and swimming at the air-water interface (Fig 5) (S4 Video). Swimming proves five times faster than crawling (21 µm/s versus 4 µm/s), since swimming amoebas do not form pseudopodia for surface adherence, and water flow with surface tension enhances their movement. This observation is similar to existing literature [23, 24]. On average, swimming amoebas exhibit two pseudopodia, while crawling amoebas display five, indicating a significant difference in locomotion strategies. This behavior may assist pathogenic amoebas like *Naegleria fowleri* in traversing from water into the nasal passages and eventually the brain [25].

**Observing complex organisms.** Trackoscope can be used to observe behaviors in complex, multicellular organisms as well; as examples, we track *Tardigrada* (*Hypsibius exemplaris*) (150 µm) and brine shrimp nauplii (*Artemia salina*) (600–1000 µm).

Long-duration tracking of *Tardigrada* in a plastic petri dish submerged in water demonstrates them taking periodic breaks across two instances for a total of 3.5 hours before moving

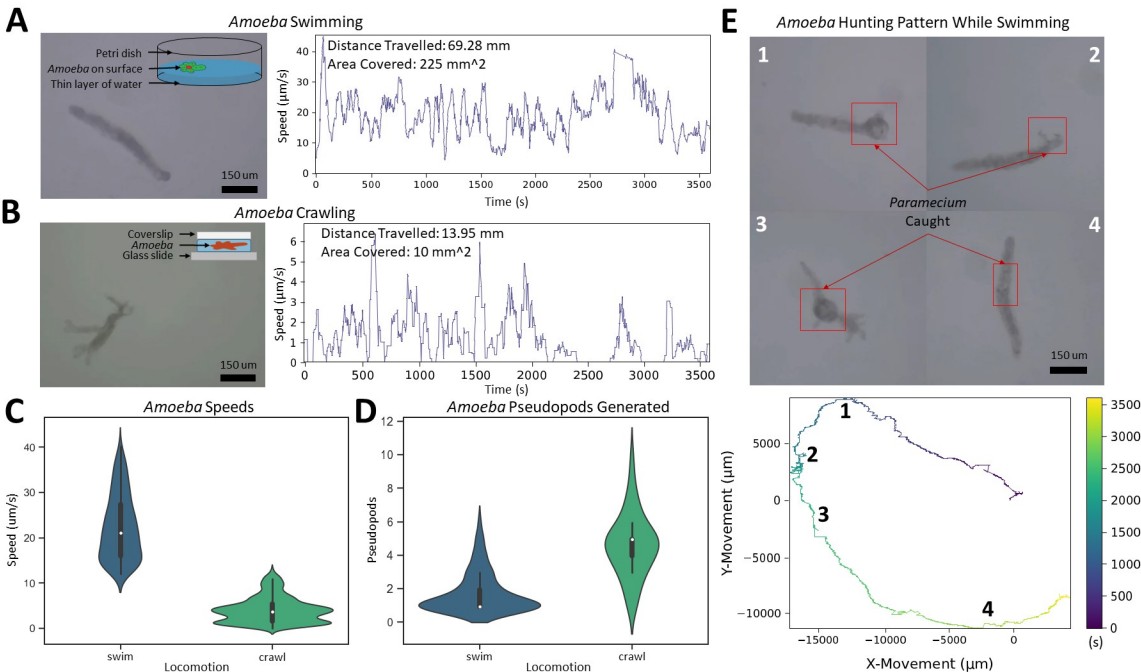

**Fig 5. Comparative movement analysis of *Amoeba proteus*: A Study of Crawling Versus Swimming Behaviors. (A)** Typical shape of *Amoeba proteus* swimming in a worm-like shape and visual of swimming setup. Track of *Amoeba proteus* swimming detailing speed and distance traveled over 1 hour. **(B)** Typical shape of *Amoeba proteus* crawling with multiple pseudopodia and visual of crawling setup. Track of *Amoeba proteus* crawling detailing speed and distance traveled over 1 hour. **(C)** Speed comparison of *Amoeba proteus* between swimming and crawling. Swimming has a higher average speed of 21 um/s while crawling has an average speed of 4 um/s. **(D)** Shape-changing tendencies based on the locomotion method through counting pseudopodia. While crawling, *Amoeba proteus* generate pseudopodia constantly while when swimming, only short pseudopodia are formed occasionally. **(E)** Pattern of a swimming *Amoeba proteus* capturing *Paramecium* over 1.5 hours (S4 Video). For this track, it was observed that every time the *Amoeba proteus* fed, it changed the general direction it was traveling in.

on and continuing to explore during the 6.5 hours of tracking (S5 Video). We analyze the *Tardigrada* walking using DeeplabCut, a markerless neural-network-based gait tracking software [26]. Enclosed in a plastic petri dish, the *Tardigrada*'s legs slip on the smooth surface while attempting a tripod walking gait (S6 Video), revealing an average speed of 40 $\mu$m/s (Fig 4c). These observed stepping distances agree with prior research [27], underscoring *Trackoscope*'s potential for facilitating new analyses of locomotion with automated pose estimation methods. The tracking also captures movement strategies such as navigating on debris.

Tracking of brine shrimp nauplii emphasizes their chaotic and random movements. In the nauplii stage, brine shrimp are still adjusting to movement with front jointed antennae and their motion is not smooth as their phyllopods (swimming appendages) have not developed yet [28]. In the track, we observe circular lurching motion [29] as the nauplii moves at an average speed of 515 $\mu$m/s with a top speed of 1200 $\mu$m/s (1.2 body lengths/second) (S6 Video). As the nauplii attempts to move linearly, it tends to move in a clockwise circle suggesting that its left antenna is stronger than the right; when coordination is achieved, fast linear motion is observed.

## *Trackoscope*'s role in enabling microorganism research in STEM

*Trackoscope*, with its economical design and tracking capabilities, empowers under-resourced laboratories and educational institutions to investigate a wide array of microorganisms, both

slow and swift. This technology surpasses traditional microscopy by documenting detailed micro-structures and comprehensive behavioral patterns. The example organism tracks demonstrate *Trackoscope*'s capability to monitor organisms at velocities from 0.1 $\mu m/s$ to 2 $mm/s$. With a vast tracking area of 325 $cm^2$, *Trackoscope* uniquely facilitates the observation of natural organism behavior, minimizing the chance of organisms colliding with container walls and altering their paths. This "pond" environment could further enable experiments examining predator-prey dynamics.

### *Trackoscope*'s customizable design for wider use

The *Trackoscope* design prioritizes both mass production and user customization. By utilizing laser-cut components for the base and motor mounts, the number of 3D-printed parts is minimized to eight, facilitating faster production. Additionally, the design allows for scaling the laser-cut parts to accommodate desired travel distances while maintaining compatibility with the majority of components.

The simple, modular approach to *Trackoscope*'s design also facilitates component modifications, enabling users to tailor the design to their specific needs. For example, acrylic components can be substituted with MDF (medium density fiberboard) to reduce costs, or the ring light and optics can be enhanced for improved imaging.

The design flexibility also extends to single-build customization. We built a version of *Trackoscope* that features a larger tracking area (625cm$^2$, size of an A4 sheet) and is constructed from hand-cut poplar wood and 3D-printed brackets instead of laser-cut acrylic (S4 Fig). In this prototype we also included a motorized Z-axis which can enable tracking and automated control in the Z-axis. This version highlights the customizability of the *Trackoscope* platform through its design design for restricted tool access and added movement capabilities via a motorized Z-axis.

### Future directions for *Trackoscope* enhancement

Looking to the future, *Trackoscope* could incorporate enhanced flexibility in tracking. One of the prototypes we constructed features a motorized Z-axis stage without Z-axis tracking due to excessive processing demands and slow performance. Introducing Z-axis tracking would enable longer observation periods, more consistently focused videos, and the collection of a third dimension of data for comprehensive analysis. Additionally, the use of economical linear rods introduces slight jerks at higher magnifications due to the actuation method. A modest investment in a more refined actuation system or reduction in platform size could mitigate this issue. Finally, a more tactile control of *Trackoscope* with a physical joystick can allow *Trackoscope* to be used as a standalone device without automated tracking capabilities and a connection to a computer running Python. This direct control will make it possible for a person to manually follow an organism while using the *Trackoscope* as a webcam.

## Materials and methods

### Construction of *Trackoscope*'s hardware framework

We designed the base parts using materials such as wood or acrylic, ensuring a straightforward construction process. To minimize costs and enhance design flexibility, we utilized 3D printing for custom parts. The system also incorporates Thorlabs' cage plates and lens tube systems, enabling users to customize the base optics design by selecting from Thorlabs' extensive range of compatible parts. We constructed the *Trackoscope* prototype, depicted in Fig 1 and S2 Fig, from laser cut quarter-inch acrylic. To demonstrate the customizability of Trackoscope, we

also built a prototype with a larger tracking area, 625cm$^2$ (size of an A4 sheet), from hand-cut lightweight poplar wood that also included a Z-axis (S4 Fig). All brackets are 3D printed using PLA filament at a 0.2 mm layer height. For a comprehensive list of materials, refer to S1 Table in S1 File. The assembly time for *Trackoscope* is approximately ninety minutes, as demonstrated in S7 Video and detailed in the assembly instructions in S1 File.

## Software architecture and user interface

We developed the firmware for the Arduino Uno using Arduino IDE. The host computer's software, written in Python, leverages libraries such as OpenCV, Matplotlib, imutils, and pyserial. We crafted the graphical user interface (GUI) with Python's Tkinter library. For converting the Raspberry Pi Zero into a webcam, we installed a camera firmware directly onto the SD card [30]. More information can be found on GitRepo (https://github.com/bhamla-lab/Trackoscope).

## Execution of organismal tracking experiments

We conducted most tracking experiments in 70 mm diameter Petri dishes, with exceptions for the *Amoeba* slide track and the *Blepharisma* and *Bursaria* tracks, which we carried out in an 8x8 cm square Petri dish. We sourced organisms from Carolina Biological Supply and maintained them at room temperature (24˚C). Before tracking experiments, we diluted the cultures in spring water. We performed the tracking experiments on a desktop computer (Precision 3630 Tower with an Intel i7–9700 CPU, Nvidia RTX 2070 GPU, and 16 GB of RAM), which achieved tracking at 120 Hz. We also conducted tracking tests on a laptop (Dell XPS 13 with an Intel i7–10710U CPU, Integrated Intel UHD Graphics, and 16 GB of RAM), reaching a tracking FPS of 60 Hz.

## Supporting information

**S1 Fig. *Trackoscope* wiring diagram.** Wiring diagram for *Trackoscope* detailing the connections between various components.
(TIF)

**S2 Fig. The *Trackoscope* prototype.** (A) CAD model of *Trackoscope* in the raised sample imaging setup. A white or contrasting covering is typically placed over the sample to create a clean background in the image. (B) *Trackoscope* is designed to be mass-produced and is constructed primarily out of laser-cut parts (acrylic or MDF) and minimal 3D printed components. It also uses standard metric nuts and bolts to join components together.
(TIF)

**S3 Fig. *Trackoscope* movement analysis calculation.** (A) ($\Delta x_{o/f}$, $\Delta y_{o/f}$) are calculated by taking the organism's location in the video frame at the start (is not always (0, 0)) and finding the displacement within the frame, $\Delta x_{o/f}(\mu m) = (\Delta x_{o/f_{previous}} + (x_{o/f_{current}} - x_{o/f_{start}})) * C_{pixels\ to\ \mu m}$, with $C_{pixels\ to\ \mu m}$ depending on the magnification. (B) ($\Delta x_p$, $\Delta y_p$) are calculated by adding up all platform displacements throughout the track; for instance, a single data point would be calculated with $\Delta x_p(\mu m) = \Delta x_{previous} + (\dot{x}_p * 50ms)$ where $\dot{x}$ is the velocity of the axis on the platform.
(TIF)

**S4 Fig. Customized *Trackoscope* prototype.** (A) The single-build customized version of *Trackoscope* featuring a motorized Z-axis and a raised sample that is observed from below. This prototype has a tracking area of 625cm$^2$ (size of an A4 sheet) and is built using limited tools (hand saw, 3D-printer, and screw-drivers). (B) The physical custom digital microscope

system used in both prototypes.
(TIF)

**S1 Video. Demonstration of *Trackoscope*.** *Trackoscope* demonstration featuring the user interface and tracking of *Blepharisma*.
(MP4)

**S2 Video. Video quality.** Video quality of *Trackoscope* viewing *Blepharisma* with a 10x objective lens with visible cilia movement. *Stentor* with a 10x objective lens with visible cilia movement and visible organelles. Two *Tardigrada* interacting with another under a 10x objective lens. And *Spirostomum* captured using a 4x objective lens with dark field illumination.
(MP4)

**S3 Video. Unicellular organism tracks.** Video of tracking binary fission of *Blepharisma* over 75 minutes, *Actinosphaerium* hunting *Paramecium* over 5.5 hours, *Bursaria* swimming, *Stentor* feeding and changing shape, and bright field and dark field illumination comparisons of *Spirostomum* swimming.
(MP4)

**S4 Video. *Amoeba proteus* locomotion.** Comparison of *Amoeba proteus* as it swims at the air-water interface and crawls on a glass slide.
(MP4)

**S5 Video. *Tardigrada* locomotion.** 6.5-hour track of a *Tardigrada* crawling around a petri dish and interacting with other *Tardigrada* and plant material.
(MP4)

**S6 Video. Multicellular Organism Tracks.** A Deeplabcut track of *Tardigrada* highlighting the tripod gait and limb recognition and a track of brine shrimp nauplii.
(MP4)

**S7 Video. Demonstration of *Trackoscope* assembly.** Assembly tutorial for *Trackoscope*.
(MP4)

**S1 File. Detailed parts list and assembly instructions (PDF).** More information can be found on GitRepo (https://github.com/bhamla-lab/Trackoscope).
(PDF)

## Acknowledgments

We thank all members of the Bhamla Lab for their feedback; Johnathan O'Neil for help setting up Deeplabcut; Alina Soifer for assistance.

## Author Contributions

**Conceptualization:** Priya Soneji.

**Data curation:** Priya Soneji.

**Formal analysis:** Priya Soneji.

**Funding acquisition:** Saad Bhamla.

**Investigation:** Priya Soneji, Elio J. Challita, Saad Bhamla.

**Methodology:** Priya Soneji, Elio J. Challita, Saad Bhamla.

**Project administration:** Saad Bhamla.

**Resources:** Saad Bhamla.

**Software:** Priya Soneji.

**Supervision:** Elio J. Challita, Saad Bhamla.

**Validation:** Priya Soneji.

**Visualization:** Priya Soneji.

**Writing – original draft:** Priya Soneji, Elio J. Challita, Saad Bhamla.

**Writing – review & editing:** Priya Soneji, Elio J. Challita, Saad Bhamla.

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
