## [Decision Letter · Decision Letter 0]

5 Mar 2024

PONE-D-23-43686Trackoscope: A Low-Cost, Open, Autonomous Tracking Microscope for Long-Term Observations of Microscale OrganismsPLOS ONE

Dear Dr. Bhamla,

Thank you for submitting your manuscript to PLOS ONE. After careful consideration, we feel that it has merit but does not fully meet PLOS ONE’s publication criteria as it currently stands. Therefore, we invite you to submit a revised version of the manuscript that addresses the points raised during the review process.

We look forward to receiving your revised manuscript.

Kind regards,

Baeckkyoung Sung, Ph.D.

Academic Editor

PLOS ONE

 [NIH Grant R35GM142588; NIGMS SEPA Grant R25GM142044; NSF Grants MCB-1817334; CAREER IOS-1941933; and the Open Philanthropy Project].  

[M.S.B. acknowledges funding support from NIH Grant R35GM142588; NIGMS SEPA

Grant R25GM142044; NSF Grants MCB-1817334; CAREER IOS-1941933; and the

Open Philanthropy Project. We thank all members of the Bhamla Lab for their

feedback; Johnathan O’Neil for help setting up Deeplabcu]

 [NIH Grant R35GM142588; NIGMS SEPA Grant R25GM142044; NSF Grants MCB-1817334; CAREER IOS-1941933; and the Open Philanthropy Project]

5. We note that Figure 1, 2ABCD, 3ABC, 4ABCD, 5ABE, S2, S4 and Trackoscope Assembly Instructions A,B,C, D AND E in your submission contain copyrighted images. All PLOS content is published under the Creative Commons Attribution License (CC BY 4.0), which means that the manuscript, images, and Supporting Information files will be freely available online, and any third party is permitted to access, download, copy, distribute, and use these materials in any way, even commercially, with proper attribution. For more information, see our copyright guidelines: http://journals.plos.org/plosone/s/licenses-and-copyright.

1. You may seek permission from the original copyright holder of Figure 1, 2ABCD, 3ABC, 4ABCD, 5ABE, S2, S4 and Trackoscope Assembly Instructions A,B,C, D AND E to publish the content specifically under the CC BY 4.0 license. 

6. Please remove your figures from within your manuscript file, leaving only the individual TIFF/EPS image files, uploaded separately. These will be automatically included in the reviewers’ PDF.

7. Thank you for uploading your study's underlying data set. Unfortunately, the repository you have noted in your Data Availability statement does not qualify as an acceptable data repository according to PLOS's standards.

Additional Editor Comments:

This manuscript can be considered for potential publication after revision with properly addressing the reviewers' comments.

Reviewers' comments:

Reviewer's Responses to Questions

**Comments to the Author**

1. Is the manuscript technically sound, and do the data support the conclusions?

Reviewer #1: Yes

Reviewer #2: Yes

Reviewer #3: Yes

2. Has the statistical analysis been performed appropriately and rigorously? 

Reviewer #1: N/A

Reviewer #2: N/A

Reviewer #3: Yes

3. Have the authors made all data underlying the findings in their manuscript fully available?

Reviewer #1: Yes

Reviewer #2: Yes

Reviewer #3: Yes

4. Is the manuscript presented in an intelligible fashion and written in standard English?

Reviewer #1: Yes

Reviewer #2: Yes

Reviewer #3: Yes

5. Review Comments to the Author

Reviewer #1: Dear Authors,

The designed system is well described in the manuscript, and all the documents uploaded to GitHub are accessible in a usable format.

Firstly, the range of the Trackoscope which is designed motorized X and Y stages with an 18 cm x 18 cm travel range, is excellent. Additionally, the micro-stepping mechanism is a very useful setup for the device. The Optical Core is designed with adequate quality, and not requiring a high-performance GPU when tracing organisms with OpenCV’s Channel and Spatial Reliability Tracker (CSRT tracker) makes the design very efficient.

In terms of the Performance Evaluation of Tracking and Imaging Capabilities, the resolution for slow-motion organisms is good. However, the resolution for fast organisms is comparatively lower than that of other systems. Nevertheless, it's important to note that there is currently no setup available, except yours, that can effectively trace these organisms at both large spatial and time scales.

Please correct the species name in the supplementary files to italic and small-capital letters.

For Trackoscope’s Customizable Design for Wider Use: I recommend adding an additional supplementary file to the manuscript, demonstrating the tracing of a multicellular organism like Botryllid ascidian (marine invertebrate) larvae or adult oral siphon movements (such as feeding behavior) or zooplankton movements. This addition could attract wider attention from the scientific community.

Reviewer #2: This is excellent paper demonstrating the design of a cheap and simply to build tracking microscope. It seems to me that a lot of thought has been given to its design and execution.

I highly recommend the publication of this manuscript.

I have one comment: The final tracks look a bit rough. The tracking dot moves around on the organism a lot. This can be fixed in post-processing I suppose? I.e. the video is saved (using Open Broadcast Studio) and then subsequent analysis could fine-tune the tracking precision. It would be nice if such a process could be demonstrated, but is definite not necessary for the publication of this manuscript. (more of a nice to have, than need to have).

Reviewer #3: Priya Soneji and authors describe Trackoscope, an automated tracking microscope enabling long-duration observations of motile microorganisms. The authors highlight limitations of conventional microscopes in studying behaviors of swimming microorganisms over extended time periods. They introduce a novel tracking microscope design utilizing low-cost, modular hardware components and custom (open) software implementing open-source computer vision algorithms. The system demonstrates impressive microscopic resolution down to 8.77 μm and adaptable speed tracking from 0.02 to 11.8 body lengths/sec across diverse organisms with unique behaviors. They limit their exploration to 2D, and discuss the complexity and cost associated with motorized Z stages-while showcasing the data accessible in these dimensions. Examples showcase insights into feeding, morphology changes, reproduction, and comparative locomotion. As an open, flexible platform made with economical parts, I believe Trackoscope promises to make automated microscopy more accessible for research and education.

Comments:

The Trackoscope design fits well into the growing ecosystem of open hardware/software instrumentation. Trackoscope provides functionality comparable to expensive commercial systems at a fraction of the cost, while utilizing open libraries such as OpenCV and others in the Python universe that enable customization without reliance on proprietary software. The integration of hardware and software components to quickly enable versatile automated tracking of cells will be useful to many researchers and educators.

The resolution analysis confirms capabilities for resolving microorganism cells and structures. Speed profiling indicates suitable performance across crawling, swimming, and ciliates with rapidly changing behavior.

Diverse organism tracking experiments highlight new behavioral insights uniquely enabled by Trackoscope's combination of resolution, field of view, and duration.

An accessible and customizable platform promotes adoption by researchers with limited budgets. Open design fosters educational uses and customization. The discussion of multiple low-cost materials such as MDF makes the project adaptable to many communities.

Minor Comments:

Additional details on tracking accuracy, effect of illumination, and computational analysis methods would further strengthen characterizations of organism behavior. As the system seems catered to ciliates, the integration of a simple dark field illumination could be uniquely beneficial for these studies with Trackoscope. This could significantly improve visualization of low-contrast transparency features in protists, like pellicles, membranes, and cytoskeletal elements. This is a minor comment however, as the system is designed with frugality in mind, and the simplicity of the illumination setup is understandable-the open and customizable optics design of Trackoscope allows flexibility in illumination methods, future iterations could investigate more imaging modes.

Figures are clear and simple, I could not find anything that requires major improvement-besides the request that the text in the plot legends be increased in size (as much as possible) for clarity in reading. The plot ticks in the amoeba data in figure 5 are especially small.

Overall Evaluation:

I recommend this manuscript for publication. The authors have designed, validated, and demonstrated a novel tracking microscope that overcomes limitations of conventional designs. Unique insights into microorganism locomotion and behaviors are enabled. The open, flexible architecture promises to expand access to automated microscopy. The integration of the many powerful python-based computer vision libraries, while discussing the benefits of simplified methods such as CSRT, are appreciated. While machine learning techniques like convolutional neural networks have revolutionized computer vision, they also have disadvantages like requiring large training datasets, extensive computational resources, and loss of interpretability. Trackoscope displays how efficient deployment of robust open-source tools can sidestep these issues and provide reliable tracking performance without complexity. Avoiding sophisticated neural networks in favor of classical CV algorithms aligns well with the overall aims of an accessible, adaptable platform. This work represents a valuable contribution with both scientific and educational impacts in the microscopy field.

6. PLOS authors have the option to publish the peer review history of their article (what does this mean?). If published, this will include your full peer review and any attached files.

Reviewer #1: **Yes: **Arzu Karahan

Reviewer #2: No

Reviewer #3: **Yes: **Adam G. Larson

---

## [Decision Letter · Decision Letter 1]

24 Jun 2024

Trackoscope: A Low-Cost, Open, Autonomous Tracking Microscope for Long-Term Observations of Microscale Organisms

PONE-D-23-43686R1

Dear Dr. Bhamla,

We’re pleased to inform you that your manuscript has been judged scientifically suitable for publication and will be formally accepted for publication once it meets all outstanding technical requirements.

Kind regards,

Baeckkyoung Sung, Ph.D.

Academic Editor

PLOS ONE

Additional Editor Comments (optional):

The reviewers' comments have been well reflected in the revised manuscript.

Reviewers' comments:

Reviewer's Responses to Questions

**Comments to the Author**

1. If the authors have adequately addressed your comments raised in a previous round of review and you feel that this manuscript is now acceptable for publication, you may indicate that here to bypass the “Comments to the Author” section, enter your conflict of interest statement in the “Confidential to Editor” section, and submit your "Accept" recommendation.

Reviewer #1: All comments have been addressed

Reviewer #2: All comments have been addressed

Reviewer #3: All comments have been addressed

2. Is the manuscript technically sound, and do the data support the conclusions?

Reviewer #1: Yes

Reviewer #2: Yes

Reviewer #3: Yes

3. Has the statistical analysis been performed appropriately and rigorously? 

Reviewer #1: Yes

Reviewer #2: N/A

Reviewer #3: Yes

4. Have the authors made all data underlying the findings in their manuscript fully available?

Reviewer #1: Yes

Reviewer #2: Yes

Reviewer #3: Yes

5. Is the manuscript presented in an intelligible fashion and written in standard English?

Reviewer #1: Yes

Reviewer #2: Yes

Reviewer #3: Yes

6. Review Comments to the Author

Reviewer #1: All my comments have been addressed in this revised version, so it can be accepted for publication.

Reviewer #2: The authors have addressed all concerns.

I recommend the paper to be published without further review.

Reviewer #3: I believe the manuscript to have made the required edits to increase clarity, as well as broader impact of the microscope. I believe the instrument would be of considerable interest and great use to those looking to study behavior of organisms both model and non-model. I believe the manuscript was clear in the original submission, and the edits to improve the quality of tracks and depth of study to have only made it better.

7. PLOS authors have the option to publish the peer review history of their article (what does this mean?). If published, this will include your full peer review and any attached files.

Reviewer #1: **Yes: **Arzu Karahan

Reviewer #2: **Yes: **Julius B. Kirkegaard

Reviewer #3: **Yes: **Adam G. Larson

---

## [Editor Report · Acceptance letter]

2 Jul 2024

PONE-D-23-43686R1 

PLOS ONE

Dear Dr. Bhamla, 

I'm pleased to inform you that your manuscript has been deemed suitable for publication in PLOS ONE. Congratulations! Your manuscript is now being handed over to our production team.

Kind regards, 

on behalf of

Dr. Baeckkyoung Sung 

Academic Editor

PLOS ONE